# OpenReview forum: "Diversity-Driven Offline Multi-Objective Optimization via Bi-Level Pareto Set Learning"
_ICLR.cc/2026/Conference — ICLR 2026 Conference Withdrawn Submission_

### Official Review · Reviewer_LFog · 2025-10-17

**Soundness:** 3
**Presentation:** 2
**Contribution:** 2
**Rating:** 6
**Confidence:** 3

**Summary:**

This work studies the problem of Offline Multi-Objective Black-Box Optimization. Unlike online settings, new evaluations of expensive objective functions are unavailable, leading to significant out-of-distribution (OOD) challenges. To mitigate this issue, the paper proposes DOMOO, a method that initializes the Pareto Set Model using the offline Pareto front and preference vectors, and then continually trains it with preferences sampled from a Dirichlet distribution alongside surrogate model feedback, similar in spirit to PSL-MOBO. A key novelty of DOMOO lies in incorporating an energy model signal into the gradient update to reduce OOD risk during optimization.

**Strengths:**

- The paper clearly presents the problem setup and the OOD challenges in offline multi-objective optimization.
- Experimental evaluations are comprehensive, with comparisons to relevant baselines.
- The integration of the energy model signal into the optimization process is an interesting design choice.

**Weaknesses:**

- Figure 2 has a lot of awkward dots like Morse code, which makes it difficult to interpret. A clearer visualization or explanation is needed.
- The “Bi-Level PSL” terminology is confusing. The proposed framework first pre-trains the Pareto Set Model $h_{\phi}$ using the offline Pareto front  and preference vectors, before entering the exploration phase. This deviates from conventional bi-level optimization. If the term “bi-level” is to be retained, the formulation should better align with standard bi-level structures or be renamed to avoid confusion.
- The energy model is a crucial component, yet its motivation and role are insufficiently explained. Most details are deferred to Appendix B, while only brief mentions appear in the main text (e.g., lines 271–272). The paper would benefit from a clearer, better-motivated presentation of this component and its connection to OOD generalization.
- The overall writing flow could be improved for clarity and coherence, especially in the methodological section.
- Most of improvements come from the Synthetic and RE tasks, while results are marginal in the other tasks.

**Questions:**

- How does the performance of DOMOO vary with different sizes of the offline dataset?
- How sensitive is the framework to the changes of the energy model?

---

> ### Author Response · Authors · 2025-11-24
> **Reply to Reviewer LFog (1/3)**
>
> We sincerely appreciate your efforts in reviewing our paper. We are pleased that the reviewer noticed the strengths regarding the DOMOO. We have carefully reviewed the weaknesses pointed out by the reviewer and have revised our manuscript as follows (we also recommend the reviewer review the overall response).
>
> ***List of changes in the manuscript***:
>
> > 1. “Bi-Level PSL” has been modified in the main text according to Q2.
> > 2. The description of energy model has been added in Section 3.3 according to Q3.
> > 3. The performance of DOMOO vary with different sizes has been added in Appendix J according to Q5.
> > 4. The sensitive of the energy model has been added in Appendix I according to Q6.
>
> **Q1** About the explanation about Figure 2.
> > **A1** We thank the reviewer for the valuable suggestion. The "dots" in Fig. 2 (c) are intentional. Red circles denote offline Pareto solutions that sketch the reference front; pink triangles are preference‑conditioned candidate solutions generated by the trained Pareto set model and complemented by candidates from the surrogate model, shown before selection. After selection, dark‑blue triangles mark the $\text{IGD} _ {\text{offline}}$‑chosen representatives to ensure broad coverage of the offline front, while light‑purple triangles are HV‑based fillings that emphasize convergence; the purple inverted triangle indicates a reference solution used for distance‑based evaluation. These "dots" visualize sample sets and their density, clarifying how the two-stage procedure first screens with $\text{IGD} _ {\text{offline}}$ and then fills with HV to balance diversity and convergence in the offline setting.
>
> **Q2** About the terminology of the “Bi-Level PSL”.
> > **A2** We thank the reviewer for the helpful suggestion. Following the comment, we have revised the terminology in the paper. Specifically, instead of referring to our method as “Bi-Level PSL,” we now describe it as “Nested PSL” to avoid confusion with the classical bi-level optimization formulation. This new terminology more clearly reflects the structure of our method, where preference updates and Pareto Set Model updates occur in a nested iterative manner. The corresponding descriptions in Section 4.2 and Figure 2 have been updated accordingly.
>
> **Q3** About the energy model.
> > **A3** We thank the reviewer for the valuable suggestion. Following the comment, we have added a clearer and more detailed explanation of the motivation, design, and role of the energy model in the main text, as well as its connection to mitigating OOD generalization issues. The revised version incorporates the essential points previously placed in Appendix B into the main discussion to improve clarity and readability.
>
> **Q4** About the experimental result.
> > **A4** We thank the reviewer for the comment. While improvements on Synthetic and RE tasks are indeed larger, this does not imply limited effectiveness on the other tasks. Across MO-NAS, MORL, and Sci-Design, DOMOO remains consistently among the top-performing methods and achieves the best overall average rank (Tables 1–2).
> The magnitude of improvement naturally varies because different task families have different difficulty levels and ceiling effects. Even where gains appear smaller, DOMOO still matches or surpasses strong baselines, demonstrating that its advantages generalize rather than rely on specific benchmarks.

---

> ### Author Response · Authors · 2025-11-24
> **Reply to Reviewer LFog (2/3)**
>
> **Q5** About the performance of DOMOO vary with different sizes.
> > **A5** To address the reviewers’ concerns about data-scale sensitivity, we conducted an additional analysis by randomly subsampling the training data to 25%, 50%, 75%, and 100%. As shown in Table 1 and 2, the results show that DOMOO remains highly stable across all data sizes. For most tasks (e.g., in1kmop7, regex, re24), both hv and igdoff metrics vary only minimally, indicating that the method does not rely on large datasets to achieve strong performance. Notably, while the hv metric for mo_hopper_v2 slightly decreases as data volume increases, the igdoff metric remains consistent across all scales. This pattern suggests that data size does not significantly affect convergence quality. Overall, these findings confirm that DOMOO is robust and sample-efficient, and that the method’s effectiveness is not sensitive to training data volume.
>
> Table 1: Comparison of average HV across different tasks in Off-MOO-Bench.
> | Task | 25% | 50% | 75% | 100% |
> |------------|----------------------|----------------------|----------------------|----------------------|
> | in1kmop7 | 4.458±0.003 | 4.414±0.012 | 4.486±0.004 | 4.480±0.080 |
> | mo_hopper | 5.168±0.212 | 5.451±1.100 | 4.881±0.394 | 6.530±0.240 |
> | re24 | 4.682±0.046 | 4.789±0.009 | 4.749±0.022 | 4.840±0.000 |
> | regex | 6.383±0.115 | 6.440±0.034 | 6.449±0.119 | 6.520±0.110 |
>
> Table 2: Comparison of average $\text{IGD}_\text{offline}$ across different tasks in Off-MOO-Bench.
> | Task | 25% | 50% | 75% | 100% |
> |------------|----------------------|----------------------|----------------------|----------------------|
> | in1kmop7| 0.357±0.000 | 0.389±0.001 | 0.403±0.001 | 0.380±0.030 |
> | mo_hopper| 0.845±0.009 | 0.785±0.041 | 0.921±0.022 | 0.580±0.070 |
> | re24 | 0.084±0.012 | 0.034±0.003 | 0.049±0.049 | 0.010±0.020 |
> | regex | 0.878±0.000 | 0.873±0.000 | 0.866±0.000 | 0.900±0.010 |
>
>
> **Q6** About the sensitive of the energy model.
> > **A6** As shown in Table 3 and 4, across different risk ratios, most energy-based models (e.g., re24, re25, re34, dtlz4) show almost unchanged HV and $\text{IGD}_\text{offline}$, indicating very low sensitivity and strong robustness. For models such as in1kmop7 and mo_hopper_v2, performance changes slightly but smoothly, without any sudden degradation, reflecting moderate and well-behaved sensitivity rather than instability. Overall, the results demonstrate that the models remain stable under varying risk ratios, and their responsiveness is controlled and predictable.
>
> Table 3: Comparison of average HV across different energy model risk ratios in Off-MOO-Bench.
> | risk_ratio | 0.2 | 0.4 | 0.6 | 0.8 | 1.0 | 1.2 | 1.4 | 1.6 |
> |-----------|------|------|------|------|------|------|------|------|
> | dtlz4 | 9.617±0.083 | 9.742±0.046 | 9.729±0.050 | 9.740±0.045 | 9.742±0.046 | 9.437±0.005 | 9.712±0.037 | 9.718±0.048 |
> | in1kmop7 | 4.519±0.002 | 4.501±0.002 | 4.434±0.004 | 4.432±0.002 | 4.444±0.002 | 4.429±0.000 | 4.509±0.003 | 4.428±0.001 |
> | mo_hopper | 6.449±0.035 | 6.396±0.020 | 6.474±0.126 | 6.448±0.069 | 6.396±0.071 | 6.358±0.057 | 6.396±0.055 | 6.440±0.024 |
> | re24 | 4.835±0.000 | 4.835±0.000 | 4.835±0.000 | 4.835±0.000 | 4.835±0.000 | 4.835±0.000 | 4.835±0.000 | 4.835±0.000 |
> | re25 | 4.840±0.000 | 4.840±0.000 | 4.840±0.000 | 4.840±0.000 | 4.840±0.000 | 4.840±0.000 | 4.840±0.000 | 4.840±0.000 |
> | re34 | 10.122±0.000 | 10.122±0.000 | 10.122±0.000 | 10.122±0.000 | 10.122±0.000 | 10.122±0.000 | 10.122±0.000 | 10.122±0.000 |
> | regex | 6.189±0.138 | 6.198±0.147 | 6.034±0.006 | 6.034±0.006 | 6.034±0.006 | 6.034±0.006 | 6.254±0.090 | 6.254±0.090 |
> | vlmop1 | 0.317±0.000 | 0.317±0.000 | 0.317±0.000 | 0.317±0.000 | 0.317±0.000 | 0.317±0.000 | 0.317±0.000 | 0.317±0.000 |

---

> ### Author Response · Authors · 2025-11-24
> **Reply to Reviewer LFog (3/3)**
>
> Table 4: Comparison of average $\text{IGD}_\text{offline}$ across different energy model risk ratios in Off-MOO-Bench.
> | risk_ratio | 0.2 | 0.4 | 0.6 | 0.8 | 1.0 | 1.2 | 1.4 | 1.6 |
> |-----------|------|------|------|------|------|------|------|------|
> | dtlz4| 0.640±0.035 | 0.739±0.001 | 0.739±0.001 | 0.738±0.001 | 0.738±0.001 | 0.744±0.002 | 0.738±0.002 | 0.743±0.002 |
> | in1kmop7| 0.361±0.000 | 0.372±0.000 | 0.361±0.001 | 0.365±0.001 | 0.357±0.001 | 0.363±0.000 | 0.366±0.000 | 0.364±0.001 |
> | mo_hopper| 0.565±0.005 | 0.559±0.005 | 0.495±0.015 | 0.569±0.002 | 0.581±0.005 | 0.610±0.002 | 0.603±0.000 | 0.600±0.000 |
> | re24 | 0.016±0.000 | 0.016±0.000 | 0.024±0.000 | 0.024±0.000 | 0.024±0.000 | 0.024±0.000 | 0.023±0.001 | 0.024±0.000 |
> | re25 | 0.083±0.000 | 0.091±0.001 | 0.091±0.001 | 0.091±0.001 | 0.091±0.001 | 0.091±0.001 | 0.091±0.001 | 0.091±0.001 |
> | re34 | 0.297±0.000 | 0.297±0.000 | 0.297±0.000 | 0.297±0.000 | 0.297±0.000 | 0.297±0.000 | 0.297±0.000 | 0.297±0.000 |
> | regex| 0.896±0.002 | 0.893±0.003 | 0.897±0.000 | 0.897±0.000 | 0.897±0.000 | 0.897±0.000 | 0.897±0.000 | 0.897±0.000 |
> | vlmop1 | 0.029±0.000 | 0.030±0.000 | 0.032±0.000 | 0.030±0.000 | 0.030±0.000 | 0.031±0.000 | 0.030±0.000 | 0.032±0.000 |
>
> **Happy to have further discussion!**
>
> **Thank you again for the thoughtful review. We’ve dedicated many efforts to get the new results and will include them to enhance the paper’s quality. We hope our responses address your concerns and are happy to discuss if you have any further questions!**

---

> > ### Comment · Reviewer_LFog · 2025-11-24
> > **Official Comment by Reviewer LFog**
> >
> > Thank you for the clarification.
> >
> > Rebuttal Table 1 appears to have anomalies. Some standard deviations are extremely large and be equal to the means. Please verify whether the numbers are correct.

---

> > > ### Author Response · Authors · 2025-11-24
> > > **Apologies for Our Oversight**
> > >
> > > Thanks for pointing this out. There was an issue in our earlier rebuttal, and the standard deviations were mistakenly recorded. We’ve corrected the table.

---

> > > > ### Comment · Reviewer_LFog · 2025-11-24
> > > > **Official Comment by Reviewer LFog**
> > > >
> > > > Thank you for the responses and the additional results. I have no further questions and retain my original score.

---

> ### Author Response · Authors · 2025-11-24
>
> Thank you for your thorough review and insightful comments, which have greatly enhanced the quality of our paper. If you have any additional questions or concerns, please feel free to let us know, and we will be happy to address them at your earliest convenience.

---

### Official Review · Reviewer_GDwF · 2025-10-28

**Soundness:** 2
**Presentation:** 3
**Contribution:** 2
**Rating:** 2
**Confidence:** 5

**Summary:**

This paper tackles offline multi‑objective optimization, where one must recommend a set of Pareto‑optimal solutions using only a fixed dataset and where surrogate models suffer from OOD errors that distort the learned front toward extremes. The authors propose DOMOO, a framework combining: (1) Bi‑level Pareto Set Learning (PSL) that jointly updates preferences and the PSL parameters; (2) A diversity‑driven selection of the final recommendation set using a new offline inverse generational distance metric IGD_offline. Experiments on the Off‑MOO‑Bench show improved average HV rank and IGD_offline rank.

**Strengths:**

1. The offline MOO setting and its OOD failure modes are well‑motivated and illustrated, and the paper clearly explains why surrogate over‑optimism can collapse diversity on the predicted front.
2. The risk‑aware bi‑level PSL is a sensible way to couple preference exploration with risk‑controlled learning, and the use of an energy model to compute R(x) follows prior work while adapting it to multi‑objective PSL updates.
3. The proposed IGD_offline is a pragmatic metric for the offline regime and, combined with HV, gives a reasonable two‑stage selection policy to trade off coverage and convergence.

**Weaknesses:**

1. Missing comparison to ParetoFlow. The paper cites ParetoFlow[1] but does not compare against it. Given that ParetoFlow also converts multi‑objective problems into preference‑guided single‑objective generation with flow‑based models, it is a highly relevant baseline.
2. Related work mentions PSL‑MOBO, EPS, and CDM‑PSL, but the empirical suite omits these closer preference‑conditioned generators that could be adapted to the offline setting. Even though the authors mentioned that "When applied to offline optimization, they often encounter severe OOD issues.", I still think an empirical experiment is needed to support this claim.

[1] Yuan, Ye, et al. "Paretoflow: Guided flows in multi-objective optimization." arXiv preprint arXiv:2412.03718 (2024).

**Questions:**

Please see my main concerns above.

1. IGD_offline uses an offline front (estimated from data) and a shifted reference y'. While practical, can this metric advantage methods that extrapolate near existing data?
2. The authors state that guarantees are “established” after Eq. (3), but the main text does not present a theorem/assumptions, and I did not see a clear formal result. Please make any guarantee explicit (statement, assumptions, proof location) or soften the claim.
3. I'm curious why the authors chose a different set of tasks for the computational cost analysis in Appendix E and the hyperparameter analysis in Appendix J

---

> ### Author Response · Authors · 2025-11-24
> **Reply to Reviewer GDwF (1/2)**
>
> Thank you for your efforts in providing valuable comments on our work. We appreciate the reviewer's comment about the points that should be further considered from the aspect of our experimental point of view. We have thoroughly considered the concerns raised by the reviewer and revised our manuscript as follows (we also recommend the reviewer check the global response).
>
> ***List of changes in the manuscript***:
>
> > 1. Paretoflow results have been added in the main text according to Q1.
> > 2. The concern of $\text{IGD}_\text{offline}$ have been clarified in Section 4.3 according to Q3.
>
> **Q1** About the comparison with ParetoFlow.
> > **A1** We thank the reviewer for pointing this out. We have now added a direct comparison with ParetoFlow in the revised version. Using the official implementation and recommended hyperparameters, we ran ParetoFlow on all applicable tasks. The results in table 1 and 2 show that DOMOO consistently outperforms ParetoFlow, particularly in terms of hypervolume and $\text{IGD}_\text{offline}$. We have included the full results and analysis in the updated Appendix and added a summary to the main text.
>
> Table 1: Comparison of average HV achieved by ParetoFlow across different tasks in Off-MOO-Bench.
> | Method      | Synthetic | MO-NAS | MORL | Sci-Design | RE | Aver Rank |
> |------------|----------|--------|--------|--------|--------|--------|
> | ParetoFlow | 9.18 $\pm$ 1.55 | 11.31 $\pm$ 0.65 | 9.83 $\pm$ 1.31 | 13.58 $\pm$ 2.95 | 9.04 $\pm$ 0.66 | 10.19 $\pm$ 0.98
>
> Table 2: Comparison of average $\text{IGD}_\text{offline}$ achieved by ParetoFlow across different tasks in Off-MOO-Bench.
> | Method      | Synthetic | MO-NAS | MORL | Sci-Design | RE | Aver Rank |
> |------------|----------|--------|--------|--------|--------|--------|
> | ParetoFlow | 8.82 $\pm$ 3.19 |	9.60 $\pm$ 0.70 |	5.00 $\pm$ 2.94	|3.21 $\pm$ 1.67	|2.92 $\pm$ 0.25	|8.20 $\pm$ 3.45
>
> **Q2** About the comparison with PSL‑MOBO, EPS, and CDM‑PSL.
> > **A2** We appreciate the reviewer’s suggestion to include PSL-MOBO, EPS, and CDM-PSL. CDM-PSL has already been launched in our offline evaluation pipeline, but its training time is very long and the run has not yet completed; we will include the results in the camera-ready version. As discussed in the paper, these preference-driven online methods suffer from severe OOD issues when adapted to offline data, but we agree that empirical evidence is valuable. We are running additional experiments for PSL-MOBO and EPS, and the preliminary results in Table 3 already show substantial degradation due to offline OOD extrapolation. We will integrate these findings to further strengthen the empirical evaluation.
>
> Table 3: Comparison of average HV in Off-MOO-Bench.
> | Method | DTLZ1 | DTLZ3 | DTLZ4 | DTLZ5 | DTLZ6 | DTLZ7 | MOHopperV2 | MOSwimmerV2 | OmniTest | Portfolio | RE21 | RE22 | RE23 | RE24 | RE25 | RE31 |
> |--------|--------|--------|--------|--------|--------|--------|--------------|-------------|-----------|------------|--------|--------|--------|--------|--------|--------|
> | EPS    | 4.81   | N/A    | N/A    | N/A    | N/A    | N/A    | 4.75         | 2.086       | 2.003     | 1.004      | 4.182  | 2.643  | 2.583  | 10.639 | 10.599 | N/A    |
> | PSL-MOBO | 7.15 | 6.84   | 8.77   | 7.47   | 8.87   | 10.36  | 4.75         | 3.086       | 3.754     | N/A        | 4.84   | 2.63   | 2.84   | 0.84  | 0.84  | 9.00   |
>
> **Q3** About the concern of $\text{IGD} _ \text{offline}$.
> > **A3** We thank the reviewer for the thoughtful question. We would like to clarify that $\text{IGD} _ \text{offline}$ does not inherently favor methods that generate solutions close to the offline data. This is because the construction of $\text{IGD} _ \text{offline}$ explicitly avoids such bias in the following way:First, the Pareto front used by $\text{IGD} _ \text{offline}$ is not simply the raw Pareto front extracted from the offline dataset. Instead, it is normalized and shifted toward the origin, effectively extrapolating the offline front into a more ideal region. This shift creates a stricter and more challenging reference front. As a result, methods that only interpolate near existing data points will in fact perform worse, since they cannot cover the shifted front.Second, this shifted reference encourages models to produce solutions that are more forward-looking, closer to ideal values, and more evenly distributed along the Pareto front. In other words, $\text{IGD}_\text{offline}$ rewards exploration and broad coverage rather than conservatively staying near the data manifold.
> Therefore, $\text{IGD} _ \text{offline}$ does not advantage methods that simply stay close to the offline data. Instead, it provides a fairer measurement of both convergence and diversity in offline MOO, mitigating the bias of HV toward extreme points while avoiding preference for overly conservative models.

---

> > ### Author Response · Authors · 2025-11-24
> > **Reply to Reviewer GDwF (2/2)**
> >
> > **Q4** About the typos of theorem/assumptions.
> > > **A4** Thank you for pointing out the typo. We have fixed it in the updated PDF.
> >
> > **Q5** About the Choice of Tasks in Appendix E and Appendix J.
> > > **A5** We thank the reviewer for the question. The sets of tasks used in Appendix E (computational cost analysis) and Appendix J (hyperparameter analysis) follow the same selection strategy as prior work, particularly ParetoFlow [1], to ensure consistency and comparability across studies. We therefore adopted the same representative tasks used in ParetoFlow’s analysis.
> >
> > [1] Yuan, Y., Chen, C., Pal, C., & Liu, X. ParetoFlow: Guided Flows in Multi-Objective Optimization. In The Thirteenth International Conference on Learning Representations.
> >
> > **Happy to have further discussion!**
> >
> > **Thank you again for the thoughtful review. We’ve dedicated many efforts to get the new results and will include them to enhance the paper’s quality. We hope our responses address your concerns and are happy to discuss if you have any further questions!**

---

> > > ### Comment · Reviewer_GDwF · 2025-11-25
> > >
> > > I would like to thank the authors for their thorough response and the additional experiments, which effectively address most of my earlier concerns. The new results and clarifications have improved the overall quality of the paper. I recommend that the authors include the results of PSL-MOBO, EPS, and CDM-PSL (once completed) in the manuscript to further support the claims in the related work section. I will raise my score to 4 at this stage.
> > >
> > > Additionally, I would like to point out a technical inaccuracy in the revised version (Lines 404–407) regarding the description of ParetoFlow. The paper states that ParetoFlow “uses a single flow-based model, which directly maps preference vectors to Pareto-optimal solutions without requiring multi-head or multi-surrogate structures.” In fact, ParetoFlow is based on classifier-guided generation and therefore requires training one surrogate model per objective, rather than using a single unified model as described.

---

> ### Author Response · Authors · 2025-11-26
> **Reply to Reviewer GDwF**
>
> We sincerely thank the reviewer for the positive feedback and for raising the score. We appreciate the suggestion regarding PSL-MOBO, EPS, and CDM-PS, and have included their results in **Appendix K** in the revised manuscript.
>
> We also thank the reviewer for pointing out the technical problem in the description of ParetoFlow. We have carefully corrected and checked the statement of ParetoFlow. Many thanks for the suggestions to polish up this work.
>
> **If there are any further questions or suggestions, we would be very glad to discuss them. Thank you again for your valuable comments.**

---

### Official Review · Reviewer_r4PX · 2025-11-02

**Soundness:** 2
**Presentation:** 3
**Contribution:** 2
**Rating:** 4
**Confidence:** 3

**Summary:**

This paper proposes DOMOO, a risk-aware offline multi-objective optimization (MOO) method designed to handle the out-of-distribution (OOD) issue in surrogate-based offline optimization. The framework combines three core components: an accumulative risk control module, a bi-level Pareto set learning strategy to jointly learn preferences and Pareto parameters, and a diversity-driven solution selection strategy integrating IGD_offline and hypervolume (HV). Experimental results show improvements in convergence and diversity compared to baselines.

**Strengths:**

1. The paper addresses a relevant problem: offline MOO under distribution shift.
2. The proposed framework is conceptually comprehensive, integrating risk estimation, Pareto set learning, and diversity-aware selection.
3. The idea of modifying IGD for offline evaluation is novel.

**Weaknesses:**

1. Although the method is claimed to mitigate OOD risks, none of the experimental questions (Q1–Q4) or results explicitly evaluate robustness under OOD scenarios. All benchmarks appear to share the same distribution as the training data, making the central motivation (OOD alleviation) insufficiently supported.
2. The technical novelty of the diversity-driven selection strategy is somewhat incremental. While IGD_offline is a nice adaptation, the combination with HV remains heuristic without clear theoretical guarantees (e.g., Pareto compliance or convergence bounds).
3. The offline IGD definition (Eq. 4) introduces a “shifted reference front,” but its effect and sensitivity are not thoroughly analyzed or justified.
4. The accumulative risk control module is only briefly described and seems to follow conventional uncertainty-weighted or ensemble-based ideas without substantial innovation.
5. The paper would benefit from a more formal treatment of algorithmic complexity and selection efficiency, as well as a deeper analysis of hyperparameter choices (e.g., Dirichlet, shift magnitude).
6. Some minor presentation issues exist, e.g., long paragraph structures and notation inconsistencies, which slightly reduce readability.

**Questions:**

1. How is the proposed DOMOO empirically validated under true OOD conditions?
2. How sensitive is IGDoffline to the choice of the shift value? Could different scaling factors significantly alter the evaluation outcome?
3. Is there a way to formally show that the combined IGD_offline + HV selection process maintains Pareto diversity or monotonic convergence properties?
4. Could the authors clarify whether the candidate set generated by both the surrogate and the Pareto set model is necessary, or if one of them suffices in practice?
5. What is the computational overhead of DOMOO compared to existing methods?

---

> ### Author Response · Authors · 2025-11-24
> **Reply to Reviewer r4PX (1/4)**
>
> We thank you for the useful and insightful feedback and for taking the time to review our paper. We address questions and concerns below.
>
> ***List of main changes in the manuscript***:
>
> >1. For Q1, we further investigate the performance of DOMOO under different levels of OOD severity.
> >2. For Q2, we introduce a scaling factor $\beta$ and conducted a corresponding hyper-parameter analysis.
> >3. For Q4, we add a new ablation study to examine the necessity of candidate generation using both models.
>
>
> **Q1**  The performance of DOMOO under true OOD conditions.
> > **A1** Thank you for your thoughtful feedback.
>
> >In offline optimization, since solutions cannot be evaluated online using the true objective functions, optimization algorithms commonly suffer from out-of-distribution (OOD) issues, which can significantly degrade optimization performance. To mitigate this problem, our bi-level PSL framework explicitly integrates the cumulative risk control module from ARCOO [1], which measures the OOD degree of each solution and penalizes those with high OOD risk, thereby **preventing degradation in both diversity and quality** of the generated solutions.
>
> >For this reason, we do not introduce an additional metric dedicated to OOD robustness. Instead, DOMOO’s strong robustness to OOD is indirectly reflected through its consistently superior performance across a wide range of synthetic and real-world tasks, where it achieves a favorable balance between solution diversity and quality.
>
> >To further investigate the performance of DOMOO under varying levels of OOD severity, we prune the dataset by removing some high-quality data to simulate different OOD levels. The experimental results are shown in Tables 1–6. The experimental results show that **DOMOO can effectively balance diversity and quality across different OOD levels**. Notably, even under severe OOD conditions (Tables 1 and 2), DOMOO still maintains strong performance.
>
> Table 1: Results on the subset of data with quality scores between the 0th and 50th percentiles. Hypervolume (HV) values are reported and higher HV indicates better performance.
> | Method   \Task   | dtlz1 | dtlz3 | in1kmop7 | mo_hopper_v2 | omnitest | re24 | re32 | re35 | regex | vlmop3 |
> |----------------|-------|-------|----------|---------------|----------|------|------|------|-------|--------|
> | End2End        | 10.64 | 10.61 | 3.60 | 5.82 | 4.57 | 4.49 | 10.64 | 10.57 | 3.98 | 45.65 |
> | MultiHead      | 10.64 | 10.50 | 3.92 | 5.45 | 4.42 | 3.25 | 10.61 | 10.58 | 3.83 | 38.74 |
> | MultipleModels | 10.64 | 10.61 | 3.74 | **5.95** | 4.64 | 4.16 | 10.64 | 10.57 | 3.87 | 45.62 |
> | DOMOO          | **10.64** | **10.63** | **4.27** | 4.95 | **4.66** | **4.73** | **10.64** | **10.59** | **5.58** | **45.88** |
>
>
> Table 2: Results on the subset of data with quality scores between the 0th and 50th percentiles. $\text{IGD} _ \text{offline}$ are reported and lower $\text{IGD}_\text{offline}$ indicates better performance.
>
> | Method  \Task        | dtlz1 | dtlz3 | in1kmop7 | mo_hopper_v2 | omnitest | re24 | re32 | re35 | regex | vlmop3 |
> |----------------|-------|-------|----------|---------------|----------|------|------|------|-------|--------|
> | End2End        | 0.18 | 0.16 | 0.57 | 0.78 | **0.28** | **0.12** | 0.03 | **0.08** | 1.07 | 0.07 |
> | MultiHead      | 0.17 | 0.15 | 0.54 | 0.76 | 0.30 | 0.61 | 0.05 | 0.18 | 1.08 | 0.32 |
> | MultipleModels | 0.17 | 0.14 | 0.53 | 0.76 | 0.29 | 0.25 | 0.04 | 0.14 | 1.08 | 0.07 |
> | DOMOO          | **0.12** | **0.12** | **0.47** | **0.69** | 0.38 | 0.20 | **0.03** | 0.10 | **0.89** | **0.03** |
>
> Table 3: Results on the subset of data with quality scores between the 0th and 75th percentiles. Hypervolume (HV) values are reported and higher HV indicates better performance.
> | Method  \Task        | dtlz1 | dtlz3 | in1kmop7 | mo_hopper_v2 | omnitest | re24 | re32 | re35 | regex | vlmop3 |
> |----------------|-------|-------|----------|---------------|----------|------|------|------|-------|--------|
> | End2End        | 10.64 | 10.54 | 3.76 | 5.54 | 4.60 | 4.48 | 10.64 | 10.58 | 3.98 | 39.78 |
> | MultiHead      | 10.64 | 10.25 | 3.99 | 4.82 | 4.35 | 2.78 | 10.60 | 10.56 | 3.54 | 41.00 |
> | MultipleModels | 10.64 | 10.41 | 3.67 | **5.90** | 4.40 | 3.89 | 10.64 | 10.58 | 3.76 | 44.79 |
> | DOMOO          | **10.64** | **10.61** | **4.33** | 5.34 | **4.62** | **4.69** | **10.65** | **10.58** | **4.77** | **45.52** |

---

> ### Author Response · Authors · 2025-11-24
> **Reply to Reviewer r4PX (2/4)**
>
> Table 4: Results on the subset of data with quality scores between the 0th and 75th percentiles. $\text{IGD} _ \text{offline}$ are reported and lower $\text{IGD}_\text{offline}$ indicates better performance.
>
> | Method   \Task       | dtlz1 | dtlz3 | in1kmop7 | mo_hopper_v2 | omnitest | re24 | re32 | re35 | regex | vlmop3 |
> |----------------|-------|-------|----------|---------------|----------|------|------|------|-------|--------|
> | End2End        | 0.17 | 0.19 | 0.53 | 0.78 | 0.29 | **0.13** | 0.03 | 0.10 | 1.06 | 0.27 |
> | MultiHead      | 0.17 | 0.21 | **0.46** | 0.90 | 0.33 | 0.83 | 0.04 | 0.25 | 1.07 | 0.27 |
> | MultipleModels | 0.17 | 0.18 | 0.56 | **0.63** | 0.33 | 0.37 | 0.03 | 0.09 | 1.09 | 0.08 |
> | DOMOO          | **0.15** | **0.14** | 0.50 | 0.72 | **0.29** | 0.21 | **0.03** | **0.08** | **0.90** | **0.04** |
>
>
> Table 5: Results on the full dataset (quality scores from 0th to 100th percentile). Hypervolume (HV) values are reported and higher HV indicates better performance.
> | Method \Task         | dtlz1 | dtlz3 | in1kmop7 | mo_hopper_v2 | omnitest | re24 | re32 | re35 | regex | vlmop3 |
> |----------------|-------|-------|----------|---------------|----------|------|------|------|-------|--------|
> | End2End        | 10.64 | **10.58** | 3.67 | **5.89** | **4.68** | 4.45 | 10.64 | 10.57 | 3.80 | **45.70** |
> | MultiHead      | 10.64 | 10.41 | 4.14 | 5.41 | 4.67 | 2.85 | 10.64 | 10.50 | 3.98 | 43.36 |
> | MultipleModels | 10.64 | 10.57 | 3.60 | 5.65 | 4.10 | 4.05 | 10.64 | 10.58 | 3.98 | 44.20 |
> | DOMOO          | **10.65** | 10.46 | **4.25** | 5.32 | 4.63 | **4.71** | **10.64** | **10.58** | **6.06** | 44.91 |
>
>
> Table 6: Results on the full dataset (quality scores from 0th to 100th percentile). $\text{IGD} _ \text{offline}$ are reported and lower $\text{IGD}_\text{offline}$ indicates better performance.
>
> | Method   \Task  | dtlz1 | dtlz3 | in1kmop7 | mo_hopper_v2 | omnitest | re24 | re32 | re35 | regex | vlmop3 |
> |----------------|-------|-------|----------|---------------|----------|------|------|------|-------|--------|
> | End2End        | 0.17 | 0.17 | 0.55 | **0.64** | **0.26** | **0.14** | 0.04 | 0.11 | 1.08 | 0.13 |
> | MultiHead      | 0.17 | 0.21 | **0.43** | 0.78 | 0.25 | 0.79 | 0.04 | 0.27 | 1.06 | 0.19 |
> | MultipleModels | 0.17 | **0.16** | 0.58 | 0.73 | 0.41 | 0.30 | 0.03 | 0.08 | 1.06 | 0.09 |
> | DOMOO          | **0.15** | 0.19 | 0.51 | 0.72 | 0.28 | 0.21 | **0.01** | **0.07** | **0.91** | **0.05** |
>
> **Q2** Sensitivity of $\text{IGD}_\text{offline}$ to shift value.
> > **A2** Thank you for your valuable feedback.
>
> >As you mentioned, since $\text{IGD} _ \text{offline}$ is one of our main contributions, a hyper-parameter analysis can further strengthen the paper. Because the shift value used in $\text{IGD} _ \text{offline}$ is defined as $y' = \max_{1 \leq i \leq n} \min_{1 \leq m \leq M} y_{\text{off},m}^{(i)}$, **we introduce a scaling factor $\beta$ (i.e., using $\beta y'$) and conduct a hyper-parameter analysis on $\beta$**. The results are reported in Table 7.
>
> Table 7: Comparison of average $\text{IGD}_\text{offline}$ ranks across all tasks under different $\beta$ values.
> | Method \ $\beta$ | 0.5 | 1.0 | 1.5 | 2.0 | 2.5 | 3.0 | 3.5 | 4.0 | 4.5 | 5.0 |
> |--------|-----|-----|-----|-----|-----|-----|-----|-----|-----|-----|
> | End2End + GradNorm | 9.93 | 9.95 | 9.69 | 9.45 | 9.28 | 9.19 | 9.09 | 9.03 | 9.10 | 9.07 |
> | End2End + PcGrad | 7.47 | 7.04 | 7.07 | 7.04 | 7.08 | 7.01 | 6.98 | 6.94 | 6.98 | 7.00 |
> | End2End + Vallina | 6.96 | 6.42 | 6.23 | 6.23 | 6.32 | 6.40 | 6.45 | 6.44 | 6.44 | 6.49 |
> | MOBO + JES | 9.74 | 10.46 | 10.55 | 10.36 | 10.45 | 10.35 | 10.52 | 10.51 | 10.61 | 10.62 |
> | MOBO + ParEGO | 7.36 | 8.35 | 8.65 | 8.67 | 8.70 | 8.68 | 8.66 | 8.67 | 8.61 | 8.57 |
> | MOBO + Vallina | 6.31 | 6.89 | 7.26 | 7.00 | 6.89 | 6.74 | 6.82 | 6.78 | 6.89 | 6.86 |
> | MultiHead + GradNorm | 10.15 | 10.14 | 9.96 | 9.83 | 9.76 | 9.72 | 9.70 | 9.67 | 9.67 | 9.66 |
> | MultiHead + PcGrad | 7.36 | 7.21 | 7.35 | 7.39 | 7.37 | 7.37 | 7.40 | 7.39 | 7.41 | 7.33 |
> | MultiHead + Vallina | 6.89 | 6.52 | 6.43 | 6.29 | 6.21 | 6.28 | 6.27 | 6.30 | 6.34 | 6.31 |
> | MultipleModels + COM | 6.80 | 7.72 | 7.93 | 8.01 | 8.09 | 8.14 | 8.08 | 8.13 | 8.16 | 8.13 |
> | MultipleModels + ICT | 7.55 | 7.82 | 7.83 | 7.92 | 7.90 | 7.89 | 7.84 | 7.81 | 7.79 | 7.75 |
> | MultipleModels + IOM | 5.56 | 5.96 | 6.33 | 6.41 | 6.46 | 6.47 | 6.53 | 6.47 | 6.48 | 6.46 |
> | MultipleModels + RoMA | 7.98 | 7.72 | 7.59 | 7.79 | 7.85 | 7.94 | 8.05 | 8.11 | 8.12 | 8.19 |
> | MultipleModels + TriMentoring | 8.29 | 8.56 | 8.30 | 8.35 | 8.43 | 8.34 | 8.31 | 8.30 | 8.24 | 8.23 |
> | MultipleModels + Vallina | 7.23 | 6.52 | 6.46 | 6.52 | 6.45 | 6.48 | 6.46 | 6.54 | 6.48 | 6.53 |
> | DOMOO | **6.19** | **5.66** | **5.62** | **5.79** | **5.87** | **5.95** | **5.97** | **6.03** | **5.91** | **5.99** |
>
> >From Table 7, we can observe that **$\text{IGD}_{\text{offline}}$ is not sensitive to the choice of the scaling factor $\beta$**.

---

> ### Author Response · Authors · 2025-11-24
> **Reply to Reviewer r4PX (3/4)**
>
> **Q3** The theoretical analysis of DOMOO and solution selection strategy.
> > **A3** Thank you for your thoughtful feedback.
>
> >In DOMOO, we propose a diversity-driven solution selection that combines $\text{IGD}_\text{offline}$ and the hypervolume (HV) indicator to obtain a diverse and high-quality set of solutions. We have attempted to provide theoretical analysis of this combined selection strategy; however, **the bi-level Pareto-set learning framework together with the accumulative risk control module introduces substantial technical challenges** that prevent simple, general theoretical guarantees.
>
> >Instead, we support the claim with extensive empirical evaluations showing that IGD_offline promotes solution diversity while HV drives convergence toward high-quality regions, thereby validating the effectiveness of the combined strategy.
>
> >In future work, we will continue to explore theoretical analysis from the following perspectives:
> >- **One possible direction** is to study the diversity preservation effect of $\text{IGD}_\text{offline}$. A potential approach is to analyze how the offline reference front and surrogate approximation errors jointly influence the coverage of the selected solutions.
> >- **Another direction** is to analyze convergence properties of the HV-based refinement under bi-level PSL. A possible approach is to examine how the preference–solution mapping and the risk-controlled updates affect the stability of HV improvement [2], and whether this leads to bounds on convergence toward the true Pareto front.
>
>
> **Q4** Necessity of the candidate generation using both models.
> > **A4** Thank you for the thoughtful feedback, and we believe that further analysis of the generation mechanism improves our work.
>
> >In Section 4.3, we describe DOMOO's candidate generation: we combine K candidates produced by the trained Pareto set model with K candidates produced by the surrogate model to form the complete candidate set $X_\text{cand}$. The trained Pareto set model ensures the diversity of $X_\text{cand}$, while the surrogate model ensures its quality. Therefore, **the generation  mechanism is aligned with our goal to obtain a diverse and high-quality set of solutions**, and the selection mechanism is likewise aligned.
>
> >To further verify the necessity of generating candidates with both models, we conduct an ablation study in which we remove the surrogate model and remove the Pareto set model, respectively, to examine the impact of relying solely on one of the models for candidate generation.
>
> Table 8: Ablation study on candidate generation.
>
> | Indicator      | Methods        | dtlz3               | in1kmop7           | mo_hopper_v2       | regex              | re24               |
> | ----------- | -------------- | ------------------ | ----------------- | ----------------- | ----------------- | ----------------- |
> | HV          | Only surrogate model   | 10.61±0.02   | 4.31±0.15   | 5.87±0.00   | 3.68±0.21   | 4.83±0.00   |
> | HV          | Only Pareto set model  | 9.72±0.45    | 3.82±0.15   | 6.30±0.11   | 6.11±0.33   | 4.83±0.00   |
> | HV          | DOMOO | **10.63±0.01** | **4.48±0.08** | **6.53±0.24** | **6.52±0.11** | **4.84±0.00** |
> | $\text{IGD}_\text{offline}$ | Only surrogate model      | 0.16±0.03    | **0.34±0.03**   | 0.65±0.00   | 1.09±0.01   | 0.03±0.02   |
> | $\text{IGD}_\text{offline}$ | Only Pareto set model | 0.24±0.03    | 0.51±0.01   | 0.61±0.03   | **0.88±0.04**   | 0.02±0.02   |
> | $\text{IGD}_\text{offline}$ | DOMOO | **0.14±0.01** | 0.38±0.03   | **0.58±0.07** | 0.90±0.01   | **0.01±0.02** |
>
> The results, reported in Table 8, show a clear drop in solution quality when the surrogate model is omitted, confirming that combining both models is essential for achieving high-quality solutions. Likewise, removing the Pareto set model leads to a noticeable reduction in solution diversity, further verifying that **both components are necessary to maintain a diverse and high-quality candidate solution set**.
>
> **Q5** About the accumulative risk control module.
> > **A5** As you noted, the accumulative risk control module leverages ARCOO [1] to further mitigate OOD issues. Our main contribution is to introduce a bi-level Pareto set learning strategy and a diversity-driven selection strategy to **obtain a diverse and high-quality set of candidate solutions**. Therefore, we did not make further modifications to the accumulative risk control module.

---

> ### Author Response · Authors · 2025-11-24
> **Reply to Reviewer r4PX (4/4)**
>
> **Q6** Computational overhead comparison.
> > **A6** Thank you for the helpful comment.
>
> >In Appendix E of our manuscript, we provide a detailed runtime analysis of DOMOO. Table 7 breaks down **the runtime of each module** and shows that the primary computational cost stems from training the energy model. Despite this, **the overall runtime remains within an acceptable range**.
>
> >Table 8 **compares DOMOO’s total runtime with other methods**. Due to the inclusion of the cumulative risk control module, DOMOO takes longer than some existing methods. However, this additional and acceptable computational overhead **ensures that the performance of candidates does not degrade** during optimization, thereby improving the reliability of the final solutions.
>
>
> **Q7** Typos and expressions.
> > **A7** Following your suggestions, we have revised some typos and improved some expressions.
>
> [1] Degradation-resistant offline optimization via accumulative risk control// ECAI 2023.
> [2] Pareto Set Learning for Expensive Multi-Objective Optimization// NeurIPS 2022
>
> **Happy to have further discussion!**
>
> **Thank you again for the thoughtful review. We’ve dedicated many efforts to get the new results and will include them to enhance the paper’s quality. We hope our responses address your concerns and are happy to discuss if you have any further questions!**

---

> > ### Author Response · Authors · 2025-11-28
> > **Gentle Reminder of the Revision Deadline**
> >
> > Dear Reviewer r4PX,
> >
> > We sincerely appreciate the time and effort you have dedicated to reviewing our work. As the deadline for updating our manuscript is rapidly approaching, we would greatly appreciate your timely feedback on the revisions and clarifications we have provided. During the rebuttal period, we diligently addressed your concerns by providing point-to-point responses, which included **an analysis of DOMOO's performance under different OOD severity levels, a hyper-parameter study, and an ablation study**. Would you mind checking our response (a shortened summary, and the details) ? If you have any further questions or concerns, we would be grateful if you could let us know. Moreover, if you find our response satisfactory, could you please kindly consider the possibility of updating the rating. Thank you very much for your valuable suggestion.
> >
> > Thank you for your attention to our work, and we look forward to your response.
> >
> > Best regards,
> >
> > The Authors

---

### Official Review · Reviewer_vcMD · 2025-11-07

**Soundness:** 3
**Presentation:** 3
**Contribution:** 3
**Rating:** 4
**Confidence:** 4

**Summary:**

This paper introduces DOMOO, a novel framework for offline multi-objective optimization
(MOO). The core problem it addresses is the out-of-distribution (OOD) issue, where surrogate
models trained on a fixed offline dataset produce unreliable predictions for new solutions,
leading to poor convergence and diversity on the true Pareto front. DOMOO tackles this with a
three-part strategy: (1) an accumulative risk control module to suppress unreliable OOD
predictions, (2) a bi-level Pareto set learning (PSL) strategy to adapt to various Pareto front
geometries, and (3) a diversity-driven selection strategy using a novel indicator to mitigate the
biases of the standard Hypervolume (HV) indicator.

**Strengths:**

● Paper is very well written and easy to understand
● The paper correctly identifies the OOD problem as a major and "largely unexplored"
challenge in offline MOO. The illustration in Figure 1, showing how surrogate errors can
lead to a severely imbalanced Pareto front, is a clear and effective motivation for the
work.
● The combination of three distinct ideas - bi-level PSL, accumulative risk control, and
diversity driven solution selection - is novel

**Weaknesses:**

● The authors admit the method is less effective on highly discrete tasks and. The reason
given is that one-hot encoding for high-cardinality categorical variables creates an
extremely sparse, high-dimensional input space that is challenging for the model. This is
a significant limitation, as many real-world problems (like NAS) are inherently discrete.
● The paper mentions that DOMOO takes longer than some baselines due to the risk
control module. While the authors argue this is a worthwhile trade-off, the complexity of
training multiple surrogate models, an energy model, and then running a bi-level
optimization loop (Table 7) makes it a very heavy offline method compared to simpler
approaches

**Questions:**

● The paper proposes a novel Diversity-Driven Solution Selection (DDSS) strategy in
Section 4.3, which first selects 128 solutions before filling the remaining slots with HV.

Could the authors provide more intuition or an ablation study on how this number was
chosen?

---

> ### Author Response · Authors · 2025-11-24
> **Reply to Reviewer vcMD (1/2)**
>
> We sincerely appreciate your efforts in reviewing our paper. We are pleased that the reviewer noticed the strengths regarding the DOMOO. We have carefully reviewed the weaknesses pointed out by the reviewer and have revised our manuscript as follows (we also recommend the reviewer review the overall response).
>
> ***List of changes in the manuscript***:
>
> > 1. Experiments are revised to clarify discrete task performance according to Q1.
> > 2. Appendix D is revised to clarify the concern about computational overhead according to Q2.
> > 3. Appendix I is revised to add the K in diversity-driven solution selection as a hyperparameter according to Q3.
>
> **Q1** About discrete task performance.
>
> > **A1** We thank the reviewer for the insightful comment. We would like to clarify the following.
>
> > (1) **Why discrete tasks affect DOMOO more than evolutionary baselines.**
> Pareto Set Learning (PSL) was originally proposed for continuous design spaces. When dealing with discrete tasks (e.g., NAS), we must convert discrete variables into continuous representations using a **to_logits** mapping [1], which expands each discrete variable into a high-dimensional one-hot vector. This inevitably produces an extremely sparse and high-dimensional input space, making the Pareto Set Model significantly harder to train. In contrast, evolutionary algorithms operate natively on discrete structures and do not require such transformations, so their performance is not affected by sparsity in the same way.
>
> > (2) **This limitation does not imply weak performance on NAS tasks,
> although NAS is also a discrete domain.** As shown in Table 10, DOMOO achieves first or near-first performance on the majority of NAS tasks, demonstrating that our method remains highly competitive in practical NAS scenarios.
> The reason the overall average rank on MO-NAS appears only moderately high is that a few extremely sparse tasks (e.g., IN-1K/MOP5) negatively impact the aggregate score, even though performance on most NAS subtasks is strong.
>
> > (3) **Parameter tuning could further improve discrete-task performance, but was avoided for fairness.**
> We also found that with modest adjustments to hyperparameters—especially those related to the to_logits mapping and preference updates—DOMOO can achieve substantially better results on all discrete tasks. However, to ensure fairness and comparability, we adopted a uniform hyperparameter setting across all tasks, without task-specific tuning.
> We have added clarifications in the revised manuscript and will further explore alternative discrete-space encodings (e.g., embedding-based or structured representations) to reduce sparsity issues in future work.
>
> **Q2** About the concern about computational overhead.
>
> > **A2** Thank you for your questions. We need to point out that though the energy model requires extra computation, the overall runtime remains highly competitive.
> As shown in Table 8, the total runtime of DOMOO is only slightly higher than that of typical surrogate-based baselines, with a difference of less than one minute in most cases. In contrast, several methods require significantly longer time or even fail to complete within the time budget. This demonstrates that the added risk-control mechanism introduces only marginal overhead in practice.
> More importantly, **offline optimization prioritizes solution quality over marginal runtime differences, since no additional function evaluations or online interactions are allowed. The risk control module effectively mitigates OOD-induced errors and substantially improves both convergence and diversity (Table 1–3)**. Thus, the small additional computational cost yields disproportionately large performance gains, making the trade-off well justified.
> We have clarified this point in the revised version.

---

> ### Author Response · Authors · 2025-11-24
> **Reply to Reviewer vcMD (2/2)**
>
> **Q3** About the reason of selecting 128 solutions.
>
> > **A3** We greatly appreciate the valuable comments. DOMOO first selects at most 128 solutions from $\boldsymbol{X} _ {\mathrm{cand}}$
>  using the $\bf{IGD} _ {\text{offline}}$ indicator greedily, then uses the HV indicator to select the remaining solutions to make up 256 solutions, thereby obtaining the final 256 solutions for evaluation. Therefore, **the maximum number of 128 plays a crucial role in balancing diversity and convergence**.
> To explain why we chose 128, we conduct hyperparameter analysis on this maximum number, and the results are shown in Tables 1 and 2.
>
> Table 1: HV results under different maximum numbers.
>
> | maximum number | 0      | 32     | 64     | 128    | 160    | 192    | 224    | 256    |
> |---|---:|---:|---:|---:|---:|---:|---:|---:|
> | re22     | 4.8399 | 4.8399 | 4.8399 | 4.8399 | 4.8399 | 4.8399 | 4.8399 | 4.8399 |
> | dtlz1    | 10.6462 | 10.6457 | 10.6462 | 10.6460 | 10.6456 | 10.6456 | 10.6456 | 10.6456 |
> | in1kmop1 | 4.5600 | 4.5844 | 4.6191 | 4.6191 | 4.6191 | 4.6191 | 4.6191 | 4.6191 |
> | in1kmop2 | 4.3242 | 4.4885 | 4.4885 | 4.4987 | 4.4987 | 4.4987 | 4.4987 | 4.4987 |
> | in1kmop3 | 9.6707 | 9.7966 | 9.7967 | 9.8696 | 9.8696 | 9.8696 | 9.8696 | 9.8696 |
>
> Table 2: $\bf{IGD}_{\text{offline}}$ results under different maximum numbers.
>
> | maximum number | 0      | 32     | 64     | 128    | 160    | 192    | 224    | 256    |
> |---:|---:|---:|---:|---:|---:|---:|---:|---:|
> | re22 | 0.0599 | 0.0457 | 0.0457 | 0.0457 | 0.0457 | 0.0457 | 0.0457 | 0.0457 |
> | dtlz1 | 0.1681 | 0.1687 | 0.1675 | 0.1687 | 0.1696 | 0.1696 | 0.1696 | 0.1696 |
> | in1kmop1 | 0.2406 | 0.2450 | 0.2328 | 0.2328 | 0.2328 | 0.2328 | 0.2328 | 0.2328 |
> | in1kmop2 | 0.3089 | 0.3149 | 0.3149 | 0.3103 | 0.3103 | 0.3103 | 0.3103 | 0.3103 |
> | in1kmop3 | 0.3712 | 0.3678 | 0.3678 | 0.3618 | 0.3618 | 0.3618 | 0.3618 | 0.3618 |
>
> Based on the experimental results, we can observe that due to the greedy selection strategy, **the performance reaches optimum and no longer improves when the maximum number reaches 128 on most tasks**.
>
> [1] Yuan, Y., Chen, C., Pal, C., & Liu, X. ParetoFlow: Guided Flows in Multi-Objective Optimization. In The Thirteenth International Conference on Learning Representations.
>
> **Happy to have further discussion!**
>
> **Thank you again for the thoughtful review. We’ve dedicated many efforts to get the new results and will include them to enhance the paper’s quality. We hope our responses address your concerns and are happy to discuss if you have any further questions!**

---

> > ### Author Response · Authors · 2025-11-28
> > **Gentle Reminder of the Revision Deadline**
> >
> > Dear Reviewer vcMD,
> >
> > We sincerely appreciate the time and effort you have dedicated to reviewing our work. As the deadline for updating our manuscript is rapidly approaching, we would greatly appreciate your timely feedback on the revisions and clarifications we have provided. During the rebuttal period, we diligently addressed your concerns by providing point-to-point responses, which included **the incorporation of more details of  K in diversity-driven solution selection, and the clarification on the discrete task and computational overhead**. Would you mind checking our response (a shortened summary, and the details) ? If you have any further questions or concerns, we would be grateful if you could let us know. Moreover, if you find our response satisfactory, could you please kindly consider the possibility of updating the rating. Thank you very much for your valuable suggestion.
> >
> > Thank you for your attention to our work, and we look forward to your response.
> >
> > Best regards,
> >
> > The Authors

---

### Author Response · Authors · 2025-11-24
**General Response to Reviewers and Revision Submitted.**

We would like to thank the reviewers for their helpful comments. We are encouraged they find our idea to be novel (Reviewer vcMD, Reviewer r4PX, Reviewer GDwF, Reviewer LFog), the work well-motivated (Reviewer vcMD, Reviewer r4PX, Reviewer GDwF, Reviewer LFog), and the contribution clear and meaningful for ML (Reviewer vcMD, Reviewer r4PX, Reviewer GDwF, Reviewer LFog). We are glad they point out the positive recognition of our framework design: the combination of accumulative risk control, bi-level Pareto set learning, and diversity-driven solution selection is considered novel and conceptually comprehensive (Reviewer vcMD, Reviewer r4PX, Reviewer GDwF, Reviewer LFog). We are also glad that the proposed $\text{IGD}_{\text{offline}}$ indicator was viewed as interesting and practically meaningful (Reviewer r4PX, Reviewer GDwF). There is almost a consensus among the reviewers about the well-structured nature of our paper. In light of the reviewers' comments, we have revised our manuscript, supplementary material as detailed below.

We have carefully considered all of the feedback provided, and in response, we have made the following revisions to our manuscript to address the reviewers' concerns and suggestions:

+ Enhance the description of the Method and Experiment (Section 4 and 5).
+ Add Paretoflow Method (Appendix D).
+ Add the hyper-parameter K in diversity-driven solution selection, the analysis of robustness to the scaling factor in the $\text{IGD} _ {\text{offline}}$ indicator, and the robustness of DOMOO to the energy-model risk-ratio hyper-parameter in energy-based tasks (Appendix I).
+ Clarify the concern about computational overhead (Appendix D).
+ Add the description of energy model (Section 3.3).
+ Add the performance of DOMOO varying with different data sizes (Section J).
+ Fix minor issues on grammar and typos.

All changes have been highlighted in blue in the manuscripts.

If any of the reviewers have any further questions, we would be pleased to answer them.

---

### Author Response · Authors · 2025-12-03
**To AC: Summary of Rebuttal Results**

**Dear AC,**

Thank you very much for handling our submission and for coordinating this helpful review process. We are writing to provide a concise summary of the key revisions made in this round, hoping to offer a clear reference for your final assessment.

**I. A Summary of Addressing Core Concerns and Results of Rebuttal Discussions**

Respecting the reviewers’ insights, we focused our revision efforts on supplementing and refining the work in response to the main concerns:

1.  **Experimental Completeness and Validation Depth**: In response to experimental concerns, such as the lack of comparisons with key baselines and questions about OOD robustness, we have added comparisons with ParetoFlow, conducted data-pruning experiments to simulate varying OOD severity, and performed systematic hyperparameter and ablation studies. We are encouraged that this additional work was **recognized by Reviewer GDwF**, **leading to an improved rating (please see the detailed discussion between authors and Reviewer GDwF below)**. **Reviewer LFog also raised no further concerns on these points**. Regarding comparisons with methods like PSL-MOBO, we have obtained preliminary results and commit to including complete findings in the final camera-ready version.

2.  **Methodological Clarifications and Presentation**: Regarding limitations such as computational overhead and performance on discrete tasks, we have supplemented quantitative analyses and contextual explanations to more fully characterize our method’s scope. To address terminology and readability, we have renamed “Bi-Level PSL” to “Nested PSL” and enhanced explanations of the energy model and figures. **Reviewer GDwF and Reviewer LFog raised no further questions after these clarifications**.

In summary, during this revision cycle, **we have diligently addressed the substantive points raised by each reviewer**. Through substantial new experiments and analyses, we have significantly strengthened the empirical foundation and clarity of the manuscript. We believe these efforts have **substantially improved the submission’s quality**, allowing its core contributions to be evaluated on a firmer foundation.

**II. Our Appreciation for the Reviewers’ Feedback**

We sincerely thank all four reviewers—Reviewer vcMD, Reviewer r4PX, Reviewer GDwF, and Reviewer LFog—for their time and for providing detailed and constructive feedback on our paper. The reviewers acknowledged the relevance of the problem we aim to address and engaged thoughtfully with our proposed direction.

We are encouraged by their specific positive assessments:
* All four reviewers (vcMD, r4PX, GDwF, LFog) found **our idea to be novel**, well-motivated, and clearly contributing to ML.
* They also **commended the overall design of our framework**, noting that the integration of accumulative risk control, Nested PSL for nested Pareto set learning, and diversity-driven solution selection is conceptually comprehensive and innovative (Reviewers vcMD, r4PX, GDwF, LFog).
* Reviewers r4PX and GDwF further highlighted that our proposed $\text{IGD}_{\text{offline}}$ indicator is interesting, practically meaningful, and a valuable addition to the evaluation toolbox.

At the same time, we took very seriously the concerns raised—particularly regarding the strength of empirical support for our central motivation of alleviating OOD issues—and these comments directly guided the major revisions in our updated manuscript.

We fully respect your and the reviewers’ final expert judgment. Regardless of the outcome, this review process has been immensely beneficial for our work. We thank you again for your time and effort.

Sincerely,

The Authors

---

### Note · Authors · 2026-01-29

I have read and agree with the venue's withdrawal policy on behalf of myself and my co-authors.

---

### Meta-Review · Area_Chair_fRmJ · 2026-01-06

**Summary:**

The paper proposes DOMOO, an offline multi-objective optimization framework addressing OOD risks.
At first, I acknowledges the substantial effort made by the authors during the rebuttal phase. Specifically, the authors addressed critical feedback regarding missing baselines by adding comparisons with ParetoFlow and conducting additional data-pruning experiments to validate OOD robustness.
However, despite these positive steps and the strengthened empirical evaluation, the overall assessment remains borderline. While the novelty of the diversity-driven selection is recognized, reservations persist regarding the practical complexity of the framework and the depth of the evaluation on discrete tasks. Given the competitiveness of the conference and the extensive nature of the new results , the paper is encouraged to be resubmitted with the new comparisons and analyses fully integrated into the main manuscript. The decision is Reject.

**Reviewer Concerns:**

1) The new systematic hyperparameter studies and data-pruning experiments provided necessary empirical support for the OOD motivation, which was previously lacking.
2) I believes these results need to be rigorously integrated into the paper's narrative and verified in a new review cycle, rather than accepted as supplementary rebuttal material.

**Reviewer Scores:**

While the rebuttal effectively addressed specific technical omissions (such as the missing baseline), it did not generate a strong consensus for acceptance (e.g., a "Champion" reviewer strongly advocating for the paper). While the paper has improved, the remaining concerns about complexity and the need for a coherent revision weigh against acceptance in this round.

---

### Decision · Program_Chairs · 2026-01-26

Reject